# Accelerating PDE-Constrained Optimization by the Derivative of Neural Operators

**Ze Cheng** [1]  **Zhuoyu Li** [1]  **Xiaoqiang Wang** [1]  **Jianing Huang** [1]  **Zhizhou Zhang** [1]  **Zhongkai Hao** [2]  **Hang Su** [2]

## Abstract

PDE-Constrained Optimization (PDECO) problems can be accelerated significantly by employing gradient-based methods with surrogate models like neural operators compared to traditional numerical solvers. However, this approach faces two key challenges: (1) **Data inefficiency**: Lack of efficient data sampling and effective training for neural operators, particularly for optimization purpose. (2) **Instability**: High risk of optimization derailment due to inaccurate neural operator predictions and gradients. To address these challenges, we propose a novel framework: (1) **Optimization-oriented training**: we leverage data from full steps of traditional optimization algorithms and employ a specialized training method for neural operators. (2) **Enhanced derivative learning**: We introduce a *Virtual-Fourier* layer to enhance derivative learning within the neural operator, a crucial aspect for gradient-based optimization. (3) **Hybrid optimization**: We implement a hybrid approach that integrates neural operators with numerical solvers, providing robust regularization for the optimization process. Our extensive experimental results demonstrate the effectiveness of our model in accurately learning operators and their derivatives. Furthermore, our hybrid optimization approach exhibits robust convergence.[1]

## 1. Introduction

In this paper, we consider solving PDE-constrained optimization problems (PDECO) in a high-dimension design

[1]Bosch (China) Invest Ltd., Shanghai, China [2]Dept. of Comp. Sci. & Techn., Institute for AI, BNRist Center, Tsinghua-Bosch Joint ML Center, Tsinghua University. Correspondence to: Ze Cheng <ze.cheng@cn.bosch.com>.

*Proceedings of the $42^{nd}$ International Conference on Machine Learning*, Vancouver, Canada. PMLR 267, 2025. Copyright 2025 by the author(s).

[1]Our code is available at https://github.com/zecheng-ai/Opt_RNO.

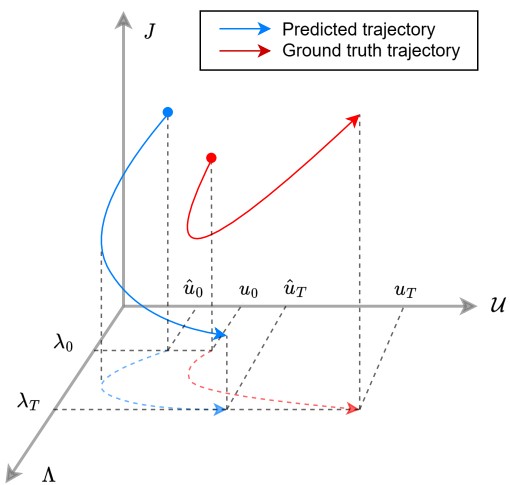

Figure 1: Illustration of Error Accumulation in Gradient-Based Optimization with Neural Operators. Blue Trajectory: Represents the optimization path $(\lambda_t, \hat{u}_t)$, where $\hat{u}_t$ is predicted by the neural operator. Red Trajectory: Represents the trajectory of the numerical solutions $u_t$ corresponding to $\lambda_t$. While the optimization process proceeds along the blue trajectory guided by the neural operator, the red trajectory of numerical solution may deviate significantly, highlighting the potential for error accumulation and suboptimal convergence.

parameter space $\Lambda$,

$$
\begin{aligned}
\min_{\lambda \in \Lambda} \quad & \tilde{J}(\lambda) \\
const. \quad & \mathcal{C}(u, \lambda) = 0
\end{aligned}
\tag{1}
$$

where $\tilde{J}(\lambda) := J(u(\lambda), \lambda)$ is the objective function, and $\mathcal{C}$ is a differential operator that describes a PDE with design parameter $\lambda$ and solution $u$. For clarity, we will henceforth omit the tilde from $J$ where the meaning is unambiguous. While gradient-free optimization methods like Bayesian optimization (Snoek et al., 2012) and particle swarm optimization (Kennedy & Eberhart, 1995) are effective for low-dimensional design spaces, gradient-based methods are generally more suitable for high-dimensional PDECO problems, for example, shape optimization (Sokolowski & Zolésio, 1992) and topology optimization (Bendsoe &

Figure 2: (Left) The overview of training neural operators with derivative learning, where curved arrow indicates differentiation by autodiff to obtain $\partial J/\partial \lambda$. (Right) The iterative process of gradient-based optimization with neural operators of fixed parameters $\theta^*$.

Sigmund, 2013). Some problems, such as those arising in density functional theory (DFT) (Kohn & Sham, 1965), may even involve a combination of both approaches at different stages.

This paper focuses on solving high-dimensional PDECO using gradient-based methods. These methods rely on the gradient of the objective function with respect to the design parameters, commonly referred to as **sensitivity**:

$$\frac{\partial \tilde{J}}{\partial \lambda} = \frac{\partial J}{\partial u}\frac{\partial u}{\partial \lambda} + \frac{\partial J}{\partial \lambda}. \tag{2}$$

The design parameters are then iteratively updated according to $\lambda_{t+1} \leftarrow \lambda_t - \eta\frac{\partial \tilde{J}}{\partial \lambda}$, where $\eta$ represents the learning rate. Among the various methods for gradient computation, the adjoint method (Pontryagin, 2018) is a well-established and efficient approach.

Despite the efficiency of the adjoint method, optimization processes can be computationally expensive due to the high cost of solving large-scale PDEs. This often results in long convergence times, spanning hours or even days. To address this, various surrogate models (Forrester et al., 2008; Zhu & Zabaras, 2018; Fahl & Sachs, 2003; Audet et al., 2000) are employed to replace computationally expensive simulations, leading to significant speedups, often reducing computation times to seconds or milliseconds. While some surrogate models directly map the design space to objective values, their applicability is limited by their dependence on the specific objective function. In contrast, surrogate models capable of predicting underlying physical fields (e.g., displacement, stress, temperature) offer greater flexibility. These models can be used to compute any desired quantity of interest, including the objective function. Neural operators (Kovachki et al., 2023; Lu et al., 2021; Hwang et al., 2022; Lu et al., 2019) exemplify this approach and show great promise for accelerating gradient-based optimization in PDECO problems.

However, training neural operators for PDECO presents significant challenges. Firstly, constructing a representative training dataset is crucial. Traditional sampling methods like Gaussian random fields (Neal, 2012) and uniform sampling often fail to capture the complex and diverse scenarios encountered in optimization problems, particularly those in-

volving intricate geometries, boundary conditions, and material properties. Moreover, effectively sampling data near the optimal region, characterized by sophisticated patterns and complex structures, using random methods is highly improbable, akin to a "monkey typing Shakespeare." (Borel, 1913)

To address these limitations, we leverage the data generated during the optimization process itself. Optimization algorithms, such as the adjoint method, produce sequences of solutions and sensitivities as they converge towards the optimal region. These sequences, often discarded as "byproducts" in traditional optimization, form valuable trajectories in the design-solution space. However, directly training neural operators on these trajectories can be inefficient and prone to overfitting due to the high correlation between successive data points. To mitigate this, we propose a training method that leverages the relative change between data points to achieve higher learning efficacy. This approach exploits the inherent structure of the optimization process, where successive iterations provide valuable information about the search direction and proximity to the optimal solution.

The second major difficulty is derailing optimization. Suppose we have a trained neural operator

$$\mathcal{G}_\theta : \Lambda \mapsto \mathcal{U} \tag{3}$$
$$\lambda \mapsto u \tag{4}$$

where $\theta$ is the learned parameter, $\Lambda$ and $\mathcal{U}$ denote design space and solution space. The gradient (2) can then be calculated by $\frac{\partial \tilde{J}}{\partial \lambda} = \frac{\partial J}{\partial u}\frac{\partial \mathcal{G}_\theta}{\partial \lambda} + \frac{\partial J}{\partial \lambda}$. Fig. 2 (right) illustrates how to compute $\frac{\partial \tilde{J}}{\partial \lambda}$ using neural operators. This approach, however, introduces significant issues: (i) There is a discrepancy between $\mathcal{G}_\theta(\lambda)$ and the true solution $u$, leading to differences between $J(\mathcal{G}_\theta)$ and $J(u)$; (ii) similarly, there exists an error between $\frac{\partial \mathcal{G}_\theta}{\partial \lambda}$ and the true derivative $\frac{\partial u}{\partial \lambda}$; (iii) through iterations of gradient-based optimization, these errors accumulate, which often results in the optimization trajectory entering out-of-distribution (OoD) regions of the neural operator, causing unreliable predictions and erratic behavior. As illustrated in Fig. 1, the error of solution prediction grows out of control from initial point $(\lambda_0, u_0)$ to the final optimized point $(\lambda_T, u_T)$. Hence, maintaining in-distribution is critical for reliable optimization.

Thus, to address the two fundamental questions, i.e., (i) how to effectively sample and utilize data near optimality, and (ii) how to control the growing error of neural operators during optimization, we proposed the following solutions:

1. We utilize the full data generated from optimization trajectories as the training set. In order to effectively achieve operator learning from such data, we adopt reference neural operators (RNO) (Cheng et al., 2024) to learn extended solution operators.

2. We analyze the effect of different neural operator architectures on their derivative performance and introduce a *Virtual-Fourier* layer to enhance derivative learning.

3. We propose a hybrid inference method that combines neural operators with numerical solvers to suppress errors and regularize optimizing process.

## 2. Related works

**Surrogate models** have received significant attention in recent years, particularly within the realm of PDECO. Early efforts such as (Audet et al., 2000) employed Kriging models (Gaussian processes) coupled with gradient-free optimization techniques. (Fahl & Sachs, 2003) applied Reduced-Order Models (ROMs) leveraging proper orthogonal decomposition (POD) alongside sophisticated optimization strategies. More recently, operator learning has emerged as a powerful paradigm, with (Hwang et al., 2022) outlining a general framework and formulating the optimization as a variational approximation. (Wang et al., 2021) explored the combination of DeepONet (Lu et al., 2019) with physics-informed loss on regular domains, while (Xue et al., 2020) demonstrated the effectiveness and versatility of amortized approaches for linear PDECO problems.

**Neural Operators**, particularly instantiated by integral operators (Kovachki et al., 2023), provide a flexible and powerful framework for modeling operators. Two prominent architectures have emerged: Fourier-based neural operator (FNO) (Li et al., 2020; Tran et al., 2021; Liu-Schiaffini et al., 2024), and transformer-based neural operators (Li et al., 2022; Cao, 2021; Hao et al., 2023; Wu et al., 2024; Xiao et al., 2024). FNOs excel on data with fixed, uniform grids by leveraging the Fast Fourier Transform. Geo-FNO (Li et al., 2023) can predict on meshes deformed from uniform grids. In contrast, transformer-based neural operators effectively handle arbitrary meshes. To enhance scalability, researchers have introduced linear transformers (Cao, 2021; Hao et al., 2023), reducing computational complexity to $O(N)$ where $N$ is the number of mesh points. Projection strategy (Wu et al., 2024) has also been proposed to decouple the complexity of attention from the number of mesh points.

Although various neural operator architectures have been explored, the impact of architecture on the smoothness of the learned operator, a crucial aspect for gradient-based optimization, remains relatively understudied. PINO (Li et al., 2024) investigates the derivatives of neural operators but focuses mainly on enforcing residual losses from governing equations. Our work delves deeper into how the architecture of neural operators influences their derivatives, essentially examining their smoothness properties.

**Sobolev training** (Czarnecki et al., 2017; Tsay, 2021) enhances the smoothness of neural networks, a critical property for gradient-based optimization. Sensitivity supervision can be viewed as a specific instance of Sobolev training. Instead of randomly sampling derivatives in all directions, sensitivity supervision focuses on derivatives with respect to the optimization variables (e.g., design parameters). DeePMD (Zhang et al., 2018) exemplifies this approach by imposing supervision on the forces acting on atoms, which are essentially the derivatives of the predicted energy with respect to the atomic positions. Our method follows a similar principle, utilizing ground truth sensitivities obtained from classical numerical methods to supervise the sensitivities predicted by the derivative of neural operators.

**Hybrid methods** combining neural networks with traditional numerical methods have shown promise in solving PDE-related problems. Notably, Hsieh et al. (2019) have demonstrated convergence guarantees for their hybrid approach, a significant advantage currently lacking in many neural operator frameworks. However, their method is limited to linear PDEs. List et al. (2022); Um et al. (2020) integrate neural networks within the numerical solution process, often refining solutions iteratively from coarse to fine grids. These existing methods focus on solving PDEs. In contrast, (Allen et al., 2022) presents a line of works on differentiable simulators. Our method is different from these works. During inference, our model cooperates with traditional solvers to enhance the accuracy and stability of the optimization process. Importantly, the neural operator is trained independently (off-line), similar to traditional neural operator approaches.

## 3. Methods

Let $d_0, d_1, h$ represent the dimension of input space, output space, and embedded space, respectively. A neural operator can then be formulated as $\mathcal{G}_\theta := \mathcal{Q} \circ \mathcal{L} \circ \cdots \circ \mathcal{L} \circ \mathcal{P}$, where $\mathcal{P} : \mathbb{R}^{d_0} \to \mathbb{R}^h$ is a lifting operator, $\mathcal{Q} : \mathbb{R}^h \to \mathbb{R}^{d_1}$ is a projection operator, and $\mathcal{L} : \mathbb{R}^h \to \mathbb{R}^h$ is the integral operator.

### 3.1. Sensitivity Loss

Since the gradient of neural operators is crucial to gradient-based optimization, we impose derivative learning to neural

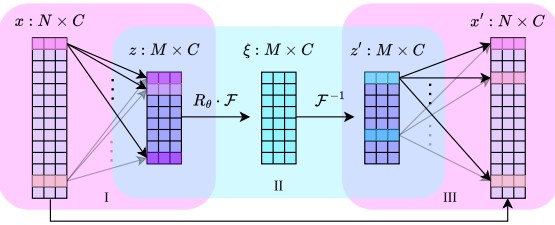

Figure 3: Overview of structure of Virtual-Fourier (VF) layer. The VF layer is composed by three steps: I. Weighted aggregating mesh points to a virtual physical space by equation (8); II. Signal processing in the virtual physical space by a Fourier-based layer (9); III. Projection of the processed signals back to physical space by equation (10).

operators. Derivative learning in this paper means imposing supervision on the sensitivity of the neural operators defined as (2). Previous studies (Czarnecki et al., 2017; Tsay, 2021; O'Leary-Roseberry et al., 2024) have shown that derivative learning is beneficial for neural networks.

Given the sensitivity $\frac{\partial J}{\partial \lambda}$ obtained from numerical solvers, along with a differentiable method of calculating $J$ given $u$ and $\lambda$, we impose a sensitivity loss on the neural operators,

$$L_{sens} = \|\nabla_\lambda J(\mathcal{G}_\theta, \lambda) - \frac{\partial J}{\partial \lambda}\|. \quad (5)$$

Fig. 2 (left) provides an overview of the computational graph for training neural operators wtih sensitivity loss. Consequently, the total loss is

$$L = \|\mathcal{G}_\theta(\lambda) - u\| + \alpha \cdot L_{sens}, \quad (6)$$

where $\alpha > 0$ is a hyperparameter controlling the weight of the sensitivity loss.

### 3.2. Virtual-Fourier Layers

The influence of neural operator architecture on derivative learning has received comparatively less attention in the literature. We find that traditional transformer-based architectures have limited expressiveness for this task. Here, we introduce a novel operator layer that exploits the simple derivative properties of Fourier layers. Notably, our layer also generalizes Fourier layers to arbitrary grids, a technical contribution that may be valuable in its own right.

In the following, we use **bold** letters to indicate tensors of different dimensions. Let us consider commonly-used transformer-based and Fourier-based neural operators. For a transformer, the derivative of the nonlinear attention unit consists of (i) the derivative of $\mathrm{Softmax}$ of attention mechanism, let $s = \mathrm{Softmax}(z)$, $z \in \mathbb{R}^N$, then $\frac{\partial s_i}{\partial z_j} = s_i \cdot (\delta_{ij} - s_j)$; (ii) the chain rule and the product rule of derivatives with respect to $Q$, $K$, and $V$. See Appendix A for more detailed analysis. The derivative of

transformers leads to limited expressiveness due to undesired inductive bias. On the other hand, the derivative of Fourier-based layers with discrete Fourier transform $\mathcal{F}$ is simply $\frac{d}{dx}\mathcal{F}(z) = \mathcal{F}(\frac{dz}{dx})$. Therefore, the expressiveness of the derivative of Fourier-based layers is untempered.

Nevertheless, Fourier-based layer has a key limitation that it can only deal with uniform mesh points of fixed numbers, while many applications, particularly optimization problems rely on irregular meshes with varying numbers of points. To address this limitation, inspired by Wu et al. (2024), we propose a *Virtual-Fourier* layer (see Fig. 3 for an overview), which transforms irregular meshes to a *virtual* physical space with a fixed number of sensors. This transformation is achieved by point-wise projection and weighted aggregation.

First, let $N$ denote the number of points in the physical space, $M$ the number of sensors in the virtual physical space, and $C$ the number of channels. While $N$ can vary across different cases, $M$ remains fixed and is pre-determined. Given input $\boldsymbol{x} \in \mathbb{R}^{N \times C}$ and a point-wise transformation $\mathrm{Project} : \mathbb{R}^{1 \times C} \to \mathbb{R}^{1 \times M}$, let

$$\boldsymbol{l}_i = \frac{\mathrm{Project}(\boldsymbol{x}_i)}{\sqrt{C}} \in \mathbb{R}^{1 \times M}. \quad (7)$$

In our implementation, $\mathrm{Project}$ is a simple linear transform, though it could potentially be implemented as a nonlinear transformation if needed. Here, $\boldsymbol{l} \in \mathbb{R}^{N \times M}$, and $l_{i,j}$ represents the logit of the $i$-th point of $\boldsymbol{x}$ being linearly classified to the $j$-th sensor in the virtual physical space. Then we apply $\mathrm{softmax}(\cdot)$ along the first dimension of $\boldsymbol{l}$ to obtain the weights $\boldsymbol{w}^{(1)} \in \mathbb{R}^{N \times M}$, specifically, $w_{i,j}^{(1)} = \exp(l_{i,j})/\sum_{i=1}^N \exp(l_{i,j})$, $i = 1, \cdots, N$. Let

$$\boldsymbol{z}_j = \sum_{i=1}^N w_{i,j}^{(1)} \boldsymbol{x}_i \in \mathbb{R}^{1 \times C}, \quad j = 1, \cdots, M. \quad (8)$$

Here, $\boldsymbol{z} \in \mathbb{R}^{M \times C}$ can be intuitively interpreted as a 1-D signal in a virtual physical space sampled by $M$ sensors, each with $C$ channels.

**Remark 3.1.** Equation (8) is similar yet different from its counterpart of Transolver (Wu et al., 2024), which let $w_i = \mathrm{Softmax}(\mathrm{Project}(x_i)) \in \mathbb{R}^{1 \times M}$ ($\mathrm{softmax}(\cdot)$ applied along the second dimension of $\boldsymbol{l}$) and $z_j = \frac{\sum_{i=1}^N w_{i,j} x_i}{\sum_{i=1}^N w_{i,j}} \in \mathbb{R}^{1 \times C}$. Both treatments preserve the nature of the whole layer as an integral operator since they only introduce additional measures on the hidden space $\mathbb{R}^C$, and the weight $\boldsymbol{w}$'s are probability density functions. The main difference lies in their derivatives. The derivative of equation (8) introduces less bias due to its simpler structure, which motivates us to adopt this form. See Appendix A for a detailed analysis of the derivatives.

These signals $z$'s are then processed by a Fourier-based layer:

$$z' = \mathcal{F}^{-1}(\boldsymbol{R}_\theta \cdot (\mathcal{F}z)) \in \mathbb{R}^{M \times C}. \quad (9)$$

The Fourier series $\mathcal{F}z$ is truncated to a finite number of modes $k \in \mathbb{Z}$ such that $\mathcal{F}z \in \mathbb{C}^{k \times C}$ and the weight parameter $\boldsymbol{R}_\theta \in \mathbb{C}^{k \times C \times C}$. Fourier-based layers assume transitional symmetry on kernel integral operator, and due to convolution theorem (Proakis, 2001), $\boldsymbol{R}_\theta$ element-wise multiplies with frequency coefficient of $z$. In the original treatment of FNO (Li et al., 2020), the interaction between channels is allowed. In this paper, with slightly abused notation, we take $1 \leq i \leq k, 1 \leq j, l \leq C$, and define the $\cdot$ operation in $(\boldsymbol{R}_\theta \cdot (\mathcal{F}z))$ from equation (9) to be $(\boldsymbol{R}_\theta \cdot (\mathcal{F}z))_{i,l} = \sum_{j=1}^{C} R_{i,l,j}(\mathcal{F}z)_{i,j}$. To see the simplicity of the derivatives of Fourier-based layer, notice $\frac{dz'}{dx} = \mathcal{F}^{-1}(\boldsymbol{R}_\theta \cdot (\mathcal{F}\frac{dz}{dx}))$, which does not change the structure of the layer. This is highly favorable for derivative learning.

Finally, the processed signals are transformed from the virtual physical space back to the physical space with the same logits from equation (7). However, this time $\mathrm{softmax}(\cdot)$ is applied on the second dimension of $l$. Namely, let $\boldsymbol{w}^{(2)} \in \mathbb{R}^{N \times M}$, and $w_{i,j}^{(2)} = \exp(l_{i,j})/\sum_{j=1}^{M} \exp(l_{i,j})$, $j = 1, \cdots, M$ and we get

$$\boldsymbol{x}_i' = \sum_{j=1}^{M} w_{i,j}^{(2)} \boldsymbol{z}_j \in \mathbb{R}^{1 \times C}. \quad (10)$$

Although Fourier-based layer is sensitive to the order of input sequence, the Virtual-Fourier layer is **permutation equivariant** similar to Transformer-based layers (without positional embedding). To see this, one needs to notice that $l_i$ only depends on $x_i$ due to the pointwise projection and the fact that under permutation equation (8) is invariant and equation (10) is equivariant.

Following (Kovachki et al., 2023), the final output of a Virtual-Fourier layer is $\sigma(W\boldsymbol{x} + b + \boldsymbol{x}')$, where $W \in \mathbb{R}^{C \times C}$, $b \in \mathbb{R}^C$ and $\sigma$ is a nonlinear activation, such as GeLU (Hendrycks & Gimpel, 2016) in our implementation.

The overall computational complexity of the layer is $\mathcal{O}(NMC + M^2C^2)$. By grouping channels into $n$ heads, the computation cost can be reduced to $\mathcal{O}(NMC + M^2C^2/n)$, and the number of parameters in $\boldsymbol{R}_\theta$ decreases from $\mathcal{O}(kC^2)$ to $\mathcal{O}(kC^2/n)$. The derivative of equation (8) and (10) do introduce some additional bias to the layer, and we provide the analysis in Appendix A.

### 3.3. Training and Optimization with RNO

Finally, a key component of our neural operator-based gradient optimization method is RNO. Its design serves two primary purposes: efficient learning from optimization trajectory data and enabling a hybrid optimization approach to mitigate error accumulation.

**Reference Neural Operators** (RNO) (Cheng et al., 2024), initially designed for learning smooth solution dependencies on domain deformations in shape optimization, can be extended to a broader range of gradient-based optimization problems. Our core idea is that RNO leverages "nearby" data points within an optimization trajectory to learn high-quality predictions. We find that the inherent relationships among these trajectory points are crucial for effective operator learning, allowing RNO to capture fine-grained solution changes resulting from small input perturbations. Specifically, our goal is to learn an operator,

$$\mathcal{G} : \Lambda \times \mathcal{U} \times \mathcal{T} \to \mathcal{U} \quad (11)$$

$$(\lambda_q, u_r, \varphi) \to u_q \quad (12)$$

where $\mathcal{U}$ and $\Lambda$ are Banach spaces, and $\mathcal{T} = C^s(\Lambda), s \geq 1$ represents a smooth transformation space on $\Lambda$. Also, we assume that there exist $\varphi \in \mathcal{T}$ such that $\varphi(\lambda_r) = \lambda_q$ and $u_r \circ \varphi^{-1} = u_q$. For example, in shape optimization, $\Lambda$ denotes the geometry space which can be substantiated as a space of signed distance function, and $\mathcal{T}$ denotes a collection of smooth deformation defined on $\Lambda$. Particularly, due to the bijection between reference mesh and query mesh, the deformation $\varphi$ can be discretized as a shift vector field between them. In topology optimization problem, $\Lambda$ represents the space of input function, $\mathcal{T}$ can be assumed as any smooth, invertible transformation space, and $\varphi$ can be substantiated as the difference $\lambda_r - \lambda_q$.

**Remark 3.2.** The well-posedness of the mapping (11), specifically the existence, uniqueness and smoothness of $\varphi$, hinges on the property of the operator $\mathcal{G}' : \Lambda \to \mathcal{U}$. These properties, such as continuity and differentiability, are non-trivial and inherently dependent on the specific PDE under consideration. For instance, the shape holomorphy of steady Stokes and Navier-Stokes equations, as demonstrated by Cohen et al. (2018), establishes the smooth dependence of the solution on small shape perturbations. This implies that given a reference shape $\lambda_r$ and a slightly perturbed shape $\lambda_q$, the difference between the corresponding solutions $u_r$ and $u_q$ is bounded by the norm of perturbation. For a comprehensive overview of shape optimization theory, we refer readers to Sokolowski & Zolésio (1992). In the context of topology optimization, analogous results of fluid flows regarding the smooth dependence of the solution on the density coefficient have been established by Haubner et al. (2023); Evgrafov (2005; 2006). These results impose certain limitations on the applicability of our method. For example, significant shape deformations that alter the topology should be avoided, and the governing PDEs must exhibit sufficient regularity with respect to the design parameters.

**Remark 3.3.** In Cheng et al. (2024), the target of RNO is

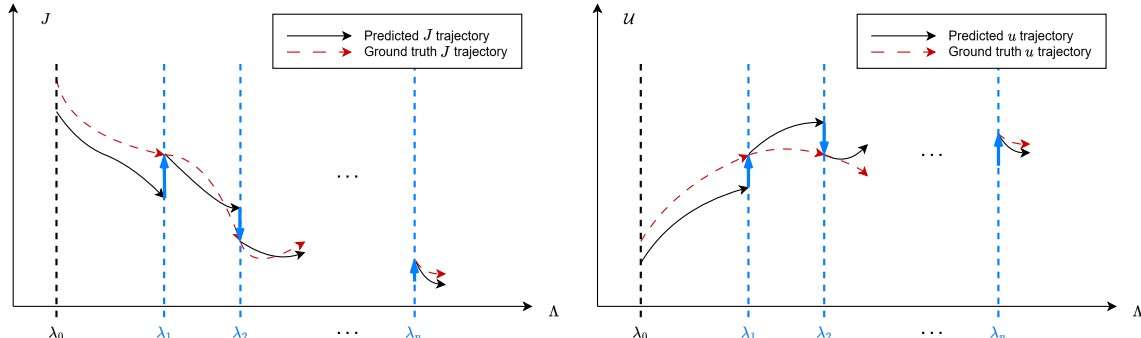

Figure 4: Optimization Process with RNO. This figure illustrates the optimization trajectory using the RNO model. Red dashed trajectory illustrates the true solutions and objectives without access. Black solid trajectory stands for optimization with neural operators. Blue arrows indicate updates to the ground truth solution based on the current input design parameters $\lambda$. The optimization trajectory can be recalibrated whenever a numerical solver is called, ensuring robustness and accuracy. (Left) The optimization trajectory is projected onto the objective function $J$ - design parameter $\Lambda$ plane. (Right) The optimization trajectory is projected onto the solution space $\mathcal{U}$ - design parameter $\Lambda$ plane. Note that both panels represent the same optimization trajectory.

defined to be the difference $\delta u = u_r \circ \varphi^{-1} - u_q$, inspired by material derivatives. We modify the target to be simply $u_q$. With skip connections outside of the integral operator layers, setting $u_q$ as the target would implicitly learn $\delta u$. There are two benefits with this modification. Firstly, it simplifies implementation. Secondly, reference neural operators can be re-interpreted as an extension of vanilla neural operator (3), which unifies RNO and NO. In practice, it enables the flexibility of utilizing RNO as NO, which is helpful if reference is expensive or less worthy acquiring.

**Remark 3.4.** Although RNO is an extension of neural operators that maps from a larger space $\Lambda \times \mathcal{U} \times \mathcal{T}$ to a solution space $\mathcal{U}$, we empirically observe that it achieves higher learning efficacy compared to vanilla neural operators. See Section 4.1. A possible underlining reason is that the training procedure of RNO is analogue to contrastive learning, which takes advantage of the inner relation between data samples, thereby improving data utilization. Given $N$ data samples, each data with $k$ neighbors, the number of pairwise data samples is $\mathcal{O}(kN)$. Since data samples are collected from optimization processes using traditional methods, they are naturally closely related. Based on this, RNO is able to effectively learn the changes of solutions that result from small changes in input.

### 3.3.1. OPTIMIZATION-ORIENTED TRAINING OF RNO

We continue to use the notation of neural operator and represent an RNO by $\mathcal{G}_\theta := \mathcal{Q} \circ \mathcal{L} \cdots \mathcal{L} \circ \sum_i^3 \mathcal{P}_i$, where $\mathcal{P}_i$'s are lifting operators corresponding to each element of triplet $(\lambda_q, u_r, \varphi)$. The training objective function is defined as (6). We summarize the training algorithm in Algorithm D. This training paradigm is specifically tailored for optimization data, thus inherently *optimization-oriented*.

**Reference-Query pairs**. To train RNO, we require a pair of data points for each query. Since our data originates from optimization trajectories, we pair each data point with its nearest neighbor within the same trajectory, excluding neighbors that exceed a predefined distance threshold. This pairing process is implemented within a custom dataloader. For further details, please refer to Appendix C.1.

**Reference dropout ratio**. During training, we randomly drop reference inputs $(u_r, \varphi)$ so that RNO reduces to a vanilla neural operator that maps from $\lambda$ to $u$. Enabling this feature is preferable because it allows RNO to perform inference flexibly without external references. This is particularly beneficial at the beginning of the optimization process, as RNO can operate without relying on any ground truth solution. Consequently, we can avoid unnecessary numerical computations during the initial stages of the design process. The reference dropout ratio is set at 0.3 to balance training performance.

### 3.3.2. OPTIMIZATION WITH RNO

To mitigate the accumulation of errors generated by neural operators during optimization, a straightforward approach involves validating intermediate optimization results and restarting the optimization process from the validated point. Then neural operators must effectively utilize newly acquired ground truth solutions. Traditional neural operators typically require retraining with the newly acquired data, which is computationally expensive in a single optimization run. In contrast, since RNO accepts a reference solution as input to predict the response for a query, it can recalibrate its predictions and adjust the optimization direction whenever a new ground-truth solution becomes available. See Fig. 4 for an illustration of the method.

| Dataset | Component | LA | PA | VF | R-VF w/o ref | R-LA | R-PA | R-VF (Ours) |
|---|---|---|---|---|---|---|---|---|
| Microreactor2D | $p$ | 7.12e-2 | 7.67e-2 | 6.36e-2 | 5.56e-2 | 1.45e-2 | 1.38e-2 | **1.33e-2** |
| | $u$ | 1.61e-1 | 1.76e-1 | 2.16e-1 | 1.25e-1 | 2.85e-2 | 2.53e-2 | **2.31e-2** |
| | $v$ | 6.87e-1 | 6.86e-1 | 6.77e-1 | 5.35e-1 | 7.34e-2 | 7.02e-2 | **6.84e-2** |
| | $c$ | 8.79e-2 | 9.33e-2 | 1.04e-1 | 7.33e-2 | 2.64e-2 | 2.56e-2 | **2.42e-2** |
| | $L_s$ | - | - | - | - | 2.05e-1 | 2.02e-1 | **1.70e-1** |
| Fuelcell2D | $p$ | 1.63e-1 | 1.18e-1 | 1.51e-1 | 1.05e-1 | 1.37e-2 | 1.34e-2 | **1.30e-2** |
| | $u$ | 2.99e-1 | 2.19e-1 | 2.97e-1 | 1.84e-1 | 1.14e-2 | 1.14e-2 | **1.10e-2** |
| | $v$ | 3.42e-1 | 2.60e-1 | 2.36e-1 | 2.10e-1 | 1.42e-2 | 1.40e-2 | **1.38e-2** |
| | $L_1$ | - | - | - | - | 4.90e-1 | 4.62e-1 | **4.44e-1** |
| | $L_2$ | - | - | - | - | 3.18e-1 | 2.77e-1 | **2.57e-1** |
| Inductor2D | $B_r$ | 5.43e-1 | 4.62e-1 | 4.57e-1 | 3.93e-1 | 9.65e-3 | 1.14e-2 | **8.66e-3** |
| | $B_z$ | 3.76e-1 | 3.24e-1 | 3.91e-1 | 3.74e-1 | 1.04e-2 | 1.05e-2 | **9.55e-3** |
| | $L_r$ | - | - | - | - | 3.12e-1 | 3.64e-1 | **3.00e-1** |
| | $L_z$ | - | - | - | - | 5.11e-1 | 5.77e-1 | **4.76e-1** |
| Drone3D | $u_1$ | 5.51e-1 | 2.42e0 | 4.27e-1 | 4.42e-1 | 1.08e-1 | 9.60e-2 | **7.64e-2** |
| | $u_2$ | 5.51e-1 | 6.60e-1 | 3.88e-1 | 3.84e-1 | 7.01e-2 | 7.25e-2 | **5.60e-2** |
| | $u_3$ | 7.50e-1 | 9.91e-1 | 1.17e0 | 5.95e-1 | 1.32e-1 | **1.18e-1** | 1.20e-1 |
| | $L_s$ | - | - | - | - | 6.48e-1 | 5.66e-1 | **5.05e-1** |

Table 1: The prefix "R-" indicates models being modified into RNO framework. See definition of baseline models in Section 4. All metrics are relative-$l_2$ error. Errors are shown by each component. $L_.$'s are sensitivity errors marked in blue. **Bold** is the best and underline is the second best. Red values are the best among all models without reference.

**Smoothen the gradient.** During inference, we adopt two tricks to further smoothen gradient while proceeding optimization: (1) Inject noise into the inputs and take average on the outputs. Let $\varepsilon_i \sim \mathcal{N}(0, \sigma)$, where $\sigma$ is set to 1% of the standard deviation of the input,

$$\delta J := \frac{1}{N_1} \sum_{i=1}^{N_1} \nabla_\lambda J(u(\lambda + \varepsilon_i), \lambda) \tag{13}$$

(2) We buffer the most recent optimization steps with certain size $M$, and then make all steps as a reference and take average on the outputs.

$$u_{pred} = \frac{1}{N_2} \sum_{j=1}^{N_2} \mathcal{G}_\theta(\lambda, u_j, \varphi_j) \tag{14}$$

The total number of forward passes for a single optimization step is $N_1 N_2$. Since the derivative of the summation of $J$ with respect to all inputs $\lambda + \varepsilon_i$ are independent to each other, the derivative in (13) can be computed by Autograd in one pass. Also, (14) can be computed in parallel. The optimization algorithm is summarized as Algorithm 1.

## 4. Experiments

We benchmark four challenging optimization datasets from various physics backgrounds, all simulated using COMSOL 6.0. For fluid dynamics we have **Microreactor2D**: A 2D topology optimization problem that maximizes the reaction rate of a channel design; **Fuelcell2D**: A 2D shape

---

**Algorithm 1** Optimization with RNO

**Input:** RNO $\mathcal{G}_\theta$, random initial input $\lambda_0$, learning rate $\eta > 0$, buffer list $B$ with size $N_2$, Ground truth solution $u_{gt} = None$ with empty initialization, $warm\_up\_steps$ for optimization before the first validation, radius $r$ around $u_{gt}$ as a criterion to trigger validation.
**for** $i = 1$ **to** $T - 1$ **do**
  **if** $i > warm\_up\_steps$ and $\text{Dist}(u_t, u_{gt}) > r$ **then**
    Update $u_{gt}$ with numerical solver
    Reset buffer $B = [u_{gt}]$
  **end if**
  Compute $u_t$ by Eq. (14) and $\delta J$ by Eq. (13)
  Store $u_t$ in $B$
  Update $\lambda_{t+1} \leftarrow \lambda_t - \eta \cdot \delta J$
**end for**

---

optimization problem that minimizes both pressure drop and flow velocity variance between channels. For eletromagnetics we have **Inductor2D**: A 3D axial-symmetric shape optimization problem that aims to reduce material cost while maximizing inductance. For solid mechanics we have **Drone3D**: A 3D topology optimization problem that minimizes elastic strain energy under two distinct load cases: vertical acceleration and motor torque. All datasets are organized as $num\_traj \times num\_step$, where $num\_traj$ and $num\_step$ are the number of trajectories and the number optimization steps. Train and test sets are split in 8:2 by trajectories. Please check more details of each dataset in Appendix. C.

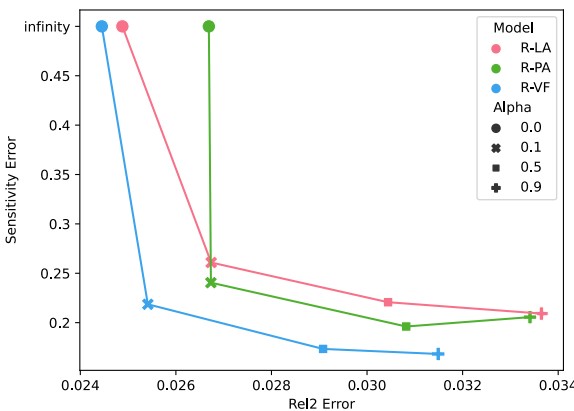

Figure 5: Trade-off between operator and derivative accuracy. This plot illustrates the trade-off for three models trained with different values of the hyperparameter $\alpha$ on the Microreactor2D dataset. R-VF consistently achieves points on the Pareto front, demonstrating a strong balance between accurate operator predictions and accurate derivative estimations.

Computing the sensitivity loss (Equation 6) requires the objective function value, $J$, for each data point. To ensure differentiability, we employ simplified approximations for the objective function. For example, we approximate the average value on irregular meshes by averaging node values and incorporate constraints as additional objectives using Lagrange multipliers. These simplifications can introduce discrepancies between the true objective function used in data generation and the approximated objective used for computing neural operator sensitivities. To address this discrepancy, we utilize an equivalent form of the cosine similarity for the sensitivity loss for all datasets except Microreactor2D, where the original formulation is used. Details of this equivalent formulation are provided in Appendix B.

We compare Virtual-Fourier (**VF**) layer to two baselines: linear attention (**LA**) from GNOT (Hao et al., 2023) and physics attention (**PA**) from Transolver (Wu et al., 2024). Except for the layer structure, all models share the same architecture, e.g., lifting and projection operators, and the number of parameters are similar. We add prefix "**R-**" to models that are modified into RNO framework.

### 4.1. Learning Results

Table 1 reveals several key observations. **Limitations of traditional training**: Vanilla neural operators trained with standard methods exhibit poor performance across all datasets. This is likely due to the inherent similarity of data points within optimization trajectories near the optimal region, leading to a lack of diversity compared to traditional training datasets. Limited diversity can easily result in overfitting.

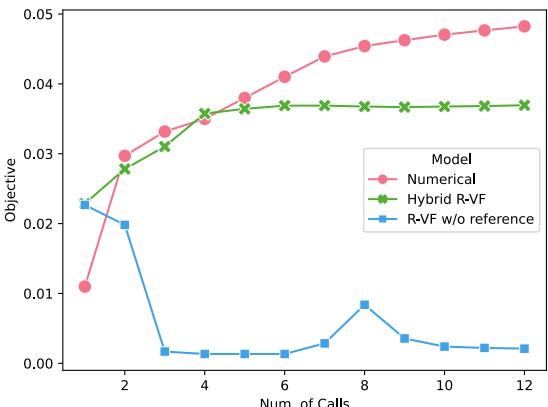

Figure 6: Objective Function Values during Optimization on Microreactor2D. The horizontal axis represents the number of calls to the numerical solver. "Hybrid R-VF" refers to the optimization algorithm (Algorithm 1) using the trained R-VF model. "R-VF w/o reference" represents the optimization process relying solely on the R-VF model without any calls to the numerical solver.

**Effectiveness of RNO**: All RNO variants demonstrate effective learning across all datasets. Notably, R-VF consistently outperforms baselines in both operator and derivative learning. We highlight that the error reduction of sensitivity loss from the second best is between 3.8% to 17.5%, suggesting that the Virtual-Fourier (VF) layer effectively reduces bias in derivative learning.

**Overfitting suppression**: Interestingly, the RNO training procedure itself appears to mitigate overfitting. When operating without a reference solution (by setting the reference dropout ratio to 1), RNO effectively degenerates into a vanilla neural operator. However, even in this degenerate state, its performance often surpasses that of all traditional neural operators, as observed in the "R-VF w/o reference" column.

#### 4.1.1. THE TRADEOFF DUE TO SENSITIVITY LOSS

The sensitivity loss (6) introduces a tradeoff between operator accuracy and derivative accuracy. To investigate this trade-off, we analyze the Pareto front of each model on Microreactor2D for different values of the coefficient $\alpha = \{0, 0.1, 0.5, 0.9\}$. Figure 5 presents the results, where the relative $l_2$ error represents the average error across all components. R-VF outperforms both baselines and demonstrates strong performance on all $\alpha$'s.

#### 4.1.2. SCALING ABILITY OF VIRTUAL-FOURIER

In order to study the scaling ability of VF, we doubled the size of Microreactor dataset to 200 trajectories (each of

12 steps and effectively 2400 samples). 40 trajectories are kept as test set. We train models on 80, 160 trajectories respectively. In Table 3 of Appendix E, we observe that with larger dataset, R-VF keeps advantage over baselines on predicting both physical fields and sensitivities.

## 4.2. Optimization Results

To evaluate our hybrid approach (Algorithm 1), we randomly selected a test case from the Microreactor2D dataset and performed optimization from scratch using a fully trained R-VF model. We employed the Method of Moving Asymptotes (MMA) (Svanberg, 1987) and Gaussian smoothing for sensitivity updates. We compared our hybrid approach with traditional numerical gradient-based optimization. Additionally, we benchmarked the R-VF model's performance within Algorithm 1 against a baseline where the R-VF model operated without feedback from the numerical solver. In this "w/o reference" scenario, we validated intermediate objective values every 20 optimization iterations.

As shown in Figure 6, the hybrid approach (R-VF with Algorithm 1) demonstrates rapid and stable convergence. In contrast, optimization relying solely on the R-VF model quickly derails without numerical solver feedback. Although the hybrid approach may converge to a suboptimal solution due to factors such as derivative accuracy errors, objective function discrepancies, constraint handling, density projection, etc., it exhibits robust convergence behavior.

Importantly, within the first four calls to the numerical solver, the hybrid R-VF optimizer achieves objective values comparable to, or even exceeding, those obtained by the numerical optimizer, suggesting potential for computational cost savings. Furthermore, each call to the numerical solver in our hybrid approach costs approximately half as much as one adjoint method step (due to the expense of solving the adjoint equations), further amplifying the potential savings.

### 4.2.1. WALL-CLOCK TIME COMPARISON

The runtime of our hybrid method consists of the time spent optimizing with RNO and the time calling numerical solvers. The latter dominates the runtime, and optimization with RNO is much cheaper. Thus, a major potential cost saving of the hybrid method lies in fewer calls for numerical solvers.

A crucial aspect of optimization with neural operators is the method of optimization. In our implementation, we adopted GD and MMA, but there are many other choices, such as Adam, L-BFGS, and SIMP (Bendsøe, 1989). A well-chosen optimization method can achieve higher objective values and reduce both the number of iterations and the number of function calls. Thus, the optimization method significantly

| | Microreactor2D | Drone3D |
|---|---|---|
| Mesh nodes | 4.9e3 | 2.1e4 |
| R-VF (GPU) | 0.53 | 0.62 |
| R-VF (CPU) | 1.89 | 3.20 |
| Numerical (CPU) | 3.20 | 49.40 |

Table 2: Wall-clock runtime comparison in $second/iter$.

affects the overall wall-clock time. Moreover, there are some important techniques in optimization, such as gradient filtering, projection, and clipping. A comprehensive and fair comparison requires all these techniques. We provide a primitive reference for wall-clock time in Table 2.

Experiments were conducted on both a laptop with CPU 2.5 GHz (11th Gen Intel i7) and a GPU Nvidia V100. The reported time is the average value of 10 iterations, measured in seconds per iteration. The runtime of traditional numerical methods increases sharply as the problem scales, since numerical methods suffer from the curse of dimensionality. In contrast, R-VF runtime increases moderately due to its linear computational complexity w.r.t. the number of mesh nodes.

## 5. Conclusion

This work addresses the challenges of applying neural operators to PDECO problems using gradient-based methods. We propose a novel framework that incorporates a specialized training procedure that leverages data generated from optimization process, a novel Virtual-Fourier layer to improve the accuracy of derivative predictions, and a hybrid approach that integrates neural operators with traditional numerical solvers. Through extensive experiments, we demonstrate the effectiveness of our proposed framework in accurately learning operators and their derivatives, leading to significant improvements in the robustness of gradient-based optimization with neural operators.

## Impact Statement

This paper presents work whose goal is to advance the field of Machine Learning. There are many potential societal consequences of our work, none which we feel must be specifically highlighted here.

## Acknowlegement

This research is a result of the collaboration within Bosch-Tsinghua Machine Learning Center. This work was supported by the National Key Research and Development Program of China (No. 2020AAA0106302), NSFC Projects (Nos. 92370124, 62350080, 92248303, 62276149).

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

## A. Derivative of Virtual-Fourier Layer

First, let us take a closer look at the derivative of nonlinear attention, $\frac{d}{dx}\text{softmax}(QK)V$, ignoring constant coefficient and the transpose of matrix. Suppose $\boldsymbol{t} \in \mathbb{R}^N$ and $\boldsymbol{s} = \text{softmax}(\boldsymbol{t})$. Then $\frac{\partial s_j}{\partial t_i} = s_j(\delta_{ij} - s_i)$. Thus, the derivative would be roughly and informally $s_j(\delta_{ij} - s_i)(Q'K + QK') + \text{softmax}(QK)V'$. Hence, the derivative of such layers is assigned with strong inductive bias (probably undesired), and the expressiveness would be severely restricted. Even for linear transformers, e.g., GNOT (Hao et al., 2023), the derivative of normalization for linear attention has a complicated form due to the quotient rule.

In contrast, the computation of the derivative of Virtual-Fourier layers essentially occurs on the Softmax function. For equation (8), its derivative is

$$\frac{\partial \boldsymbol{z}_j}{\partial \boldsymbol{x}_i} = w_{i,j}^{(1)} + \underbrace{\sum_{k=1}^{N} w_{k,j}^{(1)}(\delta_{i,k} - w_{i,j}^{(1)})\frac{\partial l_{i,j}}{\partial \boldsymbol{x}_i}\boldsymbol{x}_i}_{\text{Additional bias}} \tag{15}$$

Proof. Notice that $l_{i,j}$ only depends on $\boldsymbol{x}_i$ by (7). $w_{i,j}^{(1)} = \exp(l_{i,j})/\sum_{i=1}^{N}\exp(l_{i,j})$, $i = 1, \cdots, N$, which is $\text{softmax}(\cdot)$ along the first dimension of $l_{i,j}$. Then it directly follows from the derivative of $\text{softmax}(\cdot)$, product rule and chain rule. $\square$

In equation (10) since the dependence on $\boldsymbol{z}_j$ is linear, we mainly focus on the derivative of $w_{k,j}^{(2)}$. Similarly, we have

$$\frac{\partial w_{k,j}^{(2)}}{\partial \boldsymbol{x}_i} = w_{k,j}^{(2)}(\delta_{i,j} - w_{k,i}^{(2)})\frac{\partial l_{k,i}}{\partial \boldsymbol{x}_i} \tag{16}$$

Proof. Again, due to (7) and $w_{i,j}^{(2)} = \exp(l_{i,j})/\sum_{j=1}^{M}\exp(l_{i,j})$, $j = 1, \cdots, M$, which is $\text{softmax}(\cdot)$ along the second dimension of $l_{i,j}$. It directly follows from the derivative of $\text{softmax}(\cdot)$, product rule and chain rule. $\square$

Both the derivative of (8) and (10) introduce some bias to the derivative of operators due to the derivative of $\text{softmax}(\cdot)$, but compared to transformer-based neural operators, the derivative of Virtual-Fourier layer is considerably simpler.

## B. Sensitivity Loss

Two technical challenges arise in computing the sensitivity loss (equation 5):

**Sensitivity Representation**: For shape optimization datasets, COMSOL provides sensitivities with respect to curve parameters (e.g., coefficients of Bernstein polynomials), not directly with respect to the coordinates of mesh nodes. To simplify the comparison, we obtain the shift in node coordinates and maximize the cosine similarity between this shift and the sensitivity of the neural operator with respect to the node coordinates. Equivalently, we normalize both vectors and minimize the $l_2$ loss between them.

**Constraint Handling**: The optimization method used within COMSOL to handle constraints introduces complexities in sensitivity computation. To simplify this aspect, we incorporate constraints as additional objectives in the objective function $J$ using Lagrange multipliers. This simplification may introduce errors in the computed sensitivities. To mitigate this, we again prioritize maximizing the cosine similarity (minimize the L2 loss between normalized sensitivities), while also choosing a small value for the coefficient $\alpha$ in equation 6.

## C. Dataset and Dataloader

### C.1. Reference Dataloader

To train RNO with pairwise data as query and reference, we use a custom dataloader to select a reference point from the same optimization trajectory for each query point. Given a trajectory $\{(\lambda_i, u_i)\}_{i=0}^{N_s}$, where $N_s$ is the number of steps. the reference for the $i$-th data point is chosen from the index range $[l, r]$, where $l = \max(0, i - d)$ and $r = \min(N_s, i + d)$. Note that the reference index $j$ can be equal to $i$. Additionally, the relative difference between the reference solution $u_r$ and the query $u_q$ must be less than a threshold $d_r$. The dataloader randomly selects a reference point that satisfies both conditions. In our implementation, $d = 2$ and $d_r = 0.5$.

Because both the query and reference are selected from the same trajectory, they share a one-to-one correspondence between mesh grids. This is inherently true for data generated from topology optimization, where all meshes are identical. For shape optimization, this one-to-one correspondence is preserved after shape deformation. This consistent mesh correspondence is a key advantage of using optimization trajectories as training data. Without it, constructing appropriate transformations $\varphi$ and implementing interpolations between different meshes, as required in (Cheng et al., 2024), would be necessary.

### C.2. Microreactor

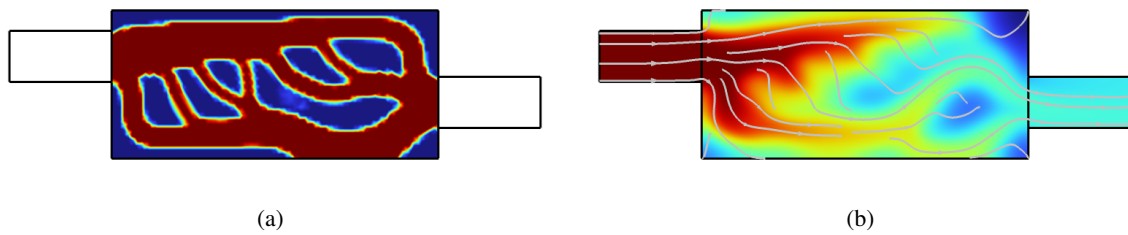

(a)                                                                                    (b)

Figure 7: (a) The optimized density $\varepsilon$ of porous media. (b) The corresponding concentration $c$.

We modified a model from COMSOL optimization gallery that designs a catalytic microreactor by topology optimization. The objective function is

$$\max_{\varepsilon} J = \frac{1}{volume(\Omega)} \int_{\Omega} k_a(1-\varepsilon)c d\Omega \tag{17}$$

where $c$ is concentration, $k_a = 1$ is the rate constant and $\varepsilon(\cdot) \in [0,1]$ denotes the control variable, the volume fraction of solid catalyst. Let $D$ be diffusion coefficient and $\mathbf{u}$ be the velocity, and the overall physics is governed by Navier-Stokes and diffusion equations,

$$\rho(\mathbf{u} \cdot \nabla)\mathbf{u} = -\nabla p + \nabla \cdot \mu(\nabla \mathbf{u} + (\nabla \mathbf{u})^T) - \alpha(\varepsilon)\mathbf{u}$$
$$\nabla \cdot \mathbf{u} = 0$$
$$\nabla \cdot (-D\nabla c) = r - \mathbf{u} \cdot \nabla c.$$

The data is structured as $\boldsymbol{\lambda} = (x_1, x_2, \varepsilon, m)$ is a 4-tensor with first two dimensions of spatial coordinates and $\varepsilon$ the control variable. The last dimension is a mask $m \in \{0,1\}^N$ for active domain where $\varepsilon$ is defined, and $N$ is the number of mesh nodes. $\mathbf{u} = (p, u, v, c, s)$, which are pressure, velocity components, concentration and the sensitivity $s = \frac{dJ}{d\varepsilon}$. Finally, let $\varphi = \varepsilon_r - \varepsilon_q$ be the transformation between reference and query.

We randomly draw the widths and heights of the reaction region, as well as the heights of inlet and outlet. The size of the dataset is $num\_traj \times num\_steps = 100 \times 12$.

### C.3. Fuelcell Bipolar Plate

We model a simple distribution area for a fuel cell bipolar plate, the governing equation is Navier-Stokes equation and the objective function consists of two parts: the total pressure drop and the variance of flow among all channels,

$$\min_{\lambda} J = (p|_{inlet} - p|_{outlet}) + \|(\mathbf{u} - \bar{\mathbf{u}})|_{channel}\|^2. \tag{18}$$

where $\lambda$ is the control variable of the internal walls in distribution areas close to inlet and outlet. The objective can be further simplified if we set $p|_{outlet} = 0$ and assume that flow velocity in each channel is constant and hence we only need to measure the velocity at end of arm area (or the inlets of channels). In COMSOL, $\lambda$ is the control variable of shapes which is modeled by the coefficients of Bernstein polynomials. In our implementation of optimizing shape with neural operators, we simply optimize the coordinates of the movable mesh nodes in arm area so that $\lambda$ stands for the coordinates of those points.

The data is structured as $\boldsymbol{\lambda} = (x_1, x_2, w, m)$, a 4-tensor with first two dimensions of spatial coordinates and $w, m \in \{0,1\}^N$ are a mask for internal walls and a mask for arm area (free shape domain), where $N$ is the number of mesh nodes.

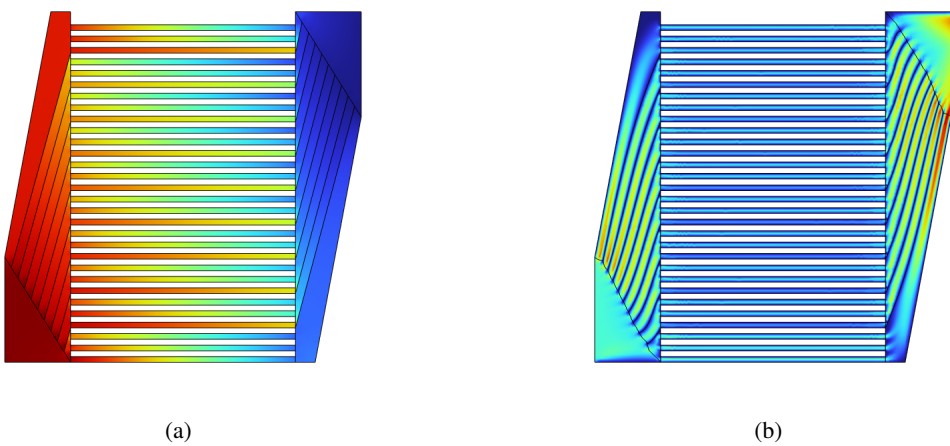

(a)     (b)

Figure 8: (a) The pressure of optimized design. (b) The velocity component $u$ of optimized design.

$\boldsymbol{u} = (p, u, v, dx_1, dx_2)$, where $(dx_1, dx_2)$ is the shift of coordinates. $\varphi = (x_1, x_2)_r - (x_1, x_2)_q$ is the difference of spatial coordinates of mesh nodes between reference and query.

We randomly draw the widths and heights of channels, as well as the widths and heights of inlet and outlet. The size of the dataset is $num\_traj \times num\_steps = 30 \times 20$.

### C.4. Inductor

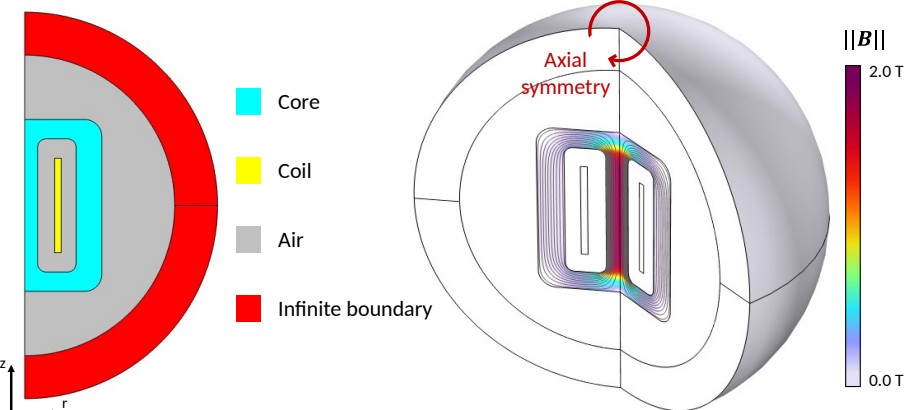

Figure 9: (Left) The 2D inductor model, consisting of core, coil, air, and infinite boundary. (Right) The complete 3D domain after revolving the domain around the symmetry axis.

We model the magnetic field response of an inductor under axial symmetry assumption as seen in Fig. 9. The objective function consists of two competing components: maximizing the inductance (magnetic energy storage capability) while minimizing the cross sectional area of the inductor,

$$\min_{\lambda} J = -\gamma \frac{\int_{\Omega} (\mathbf{B} \cdot \mathbf{H}) 2\pi r dr dz}{I^2} + (1 - \gamma) \int_{\Omega} 1 dr dz \tag{19}$$

where $\mathbf{H}$ stands for magnetic field intensity, $\mathbf{B}$ for magnetic flux density, $I$ for current, and $\lambda$ for the control variable for the cross section shape. Particularly, we randomly draw $\gamma \in [0.7, 0.9]$ to enable diversity on objective functions. For simplicity, we investigate the steady state frequency domain response at $f = \frac{\omega}{2\pi} = 1$ kHz by solving the magnetic wave equation in the 2D inductor cross section $\Omega$ using COMSOL,

$$\nabla^2 \mathbf{H} + \omega^2 \epsilon \mu \mathbf{H} = -\nabla \times \mathbf{J} \tag{20}$$

where $\mathbf{J}$ stands for current density, $\omega$ is the angular frequency, $\mu$ and $\epsilon$ are permeability and permittivity. The excitation current is set to have a small amplitude so that the induced magnetic field can be assumed to stay within the linear regime ($\mu = 600$ Tm/A) of the core material without hysteresis effect. Therefore, the steady state field response takes the form $(\mathbf{B}, \mathbf{H}) = (\tilde{\mathbf{B}}, \tilde{\mathbf{H}})e^{i\omega t}$ at identical phase. The objective function in Eq. 19 can then be simplified by replacing $\mathbf{B} \cdot \mathbf{H}$ with the corresponding amplitude $\mu\tilde{\mathbf{H}}^2$ and performing integral only within the core region, as $\mu$ is only significant within the core region.

The data is structured as $\boldsymbol{\lambda} = (r, z, \gamma, m_1, m_2)$, a 5-tensor with first two dimensions of axial-symmetrical spatial coordinates $(r, z)$ and $m_1, m_2 \in \{0, 1\}^N$ are masks for coils and magnetic core, where $N$ is the number of mesh nodes. $\boldsymbol{u} = (B_r, B_z, dr, dz)$, where $(dr, dz)$ are shifts of coordinates. $\varphi = (r, z)_r - (r, z)_q$ is the difference of spatial coordinates of mesh nodes between reference and query.

We randomly draw the radius and heights of core, the radius and heights of inner hole, the widths and heights of coil and the parameter $\gamma$ in objective function. The size of the dataset is $num\_traj \times num\_steps = 100 \times 10$.

### C.5. Drone

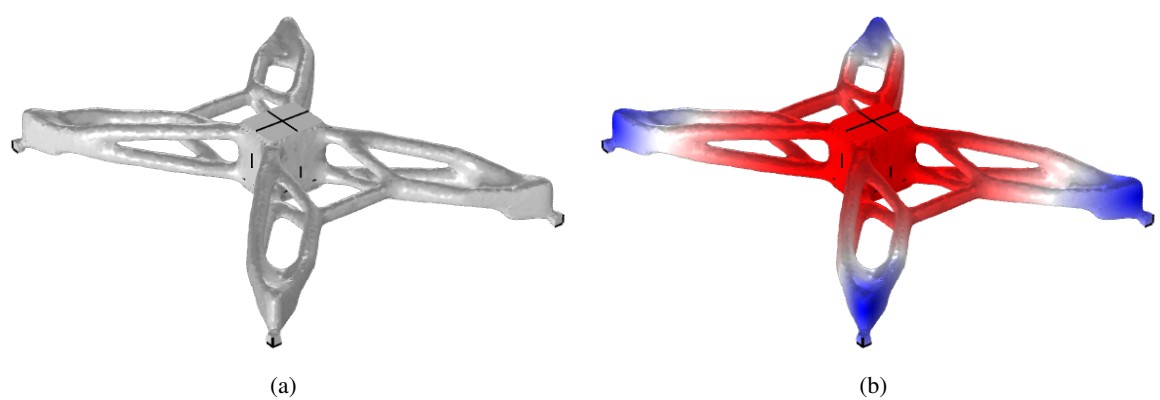

(a)                                                               (b)

Figure 10: (a) The optimized density $\varepsilon$ filtered by a threshold. (b) The magnitude of corresponding displacement $\boldsymbol{u}$ of the loadcase of vertical acceleration.

We modified a model from COMSOL optimization gallery that designs the arms of a drone with linear elastic material. The problem minimizes elastic strain energy with physical variables such as $3 \times 3$ strain, stress tensors and displacement. Our modification on the objetive is, with Einstein sum notation,

$$\min_{\varepsilon} J = \int_V b_i u_i dV + \int_{\partial R} t_i u_i dA, \tag{21}$$

where $u_i$ is the displacement, $b_i$ is force, $t_i$ is traction on surface $\partial R$ and $\varepsilon$ denotes the density of the plastic material. This is equivalent to elastic strain energy due to the principle of minimum potential energy (Bower, 2009). Also, the governing equation will be simplified as the Navier equations of elasticity (Chapter 5. Bower (2009)). The only field variable of interest is displacement. The problem is optimized under two load cases, a vertical acceleration on the whole body and a torque on the motor surface. The optimization process is performed on a single arm, and the final result is then mirrored twice to obtain the complete drone body. See Fig. 10.

The data is structured as $\boldsymbol{\lambda} = (x_1, x_2, x_3, \theta, f_1, f_2, f_3, m_1, m_2, m_3)$, a 10-tensor with first 3 spatial coordinates and $\theta \in [0, 1]^N$ the control variable of density model for topology optimization where $N$ is the number of mesh nodes. $f$'s are force components and $m$'s are masks for the surface where torque is applied, surfaces for symmetry conditions and volume assigned with $\theta$. Displacement $\boldsymbol{u} = (u_1, u_2, u_3, s)$, sensitivity $s = \frac{\partial J}{\partial \theta}$, and transformation $\varphi = \boldsymbol{\theta}_r - \boldsymbol{\theta}_q$.

We randomly draw the widths, heights and thickness of the arm of drones, as well as the torque and accelerations of of two loadcases. The size of the dataset is $num\_traj \times num\_steps = 10 \times 20$.

### D. Algorithms

---

**Algorithm 2** Optimization-oriented training of RNO

---

**Input:** RNO $\mathcal{G}_\theta$, dataloader that loads data in pairs, $(\lambda_q, u_q)$ and $(\lambda_r, u_r)$, flag $Sens$ for training with sensitivity loss, dropout ratio $r_{drop}$ that drops reference.

**for** $epoch = 0$ **to** $N_e - 1$ **do**

  Draw $a \sim \mathcal{U}[0, 1]$

  **if** $a < r_{drop}$ **then**

    Drop $u_r$ and $\varphi$

  **end if**

  $u_{pred} = \mathcal{G}_\theta(\lambda_q, u_r, \varphi)$

  Compute loss $L = \|u_{pred} - u_q\|$

  **if** Sens **then**

    Compute $J$ and $\frac{\partial J}{\partial \lambda_q}$ by autodiff

    Update loss $L$ by (6).

  **end if**

  $optimizer.zero()$

  $L.backward()$

  $optimizer.update()$

**end for**

---

# E. More Experiments

| | $p$ | $u$ | $v$ | $c$ | $L_s$ |
|---|---|---|---|---|---|
| R-LA-80 | 1.64e-2 | 3.25e-2 | 7.83e-2 | 2.65e-2 | 2. 69e-2 |
| R-PA-80 | 1.51e-2 | 2.74e-2 | **7.46e-2** | 2.48e-2 | 2.54e-2 |
| R-VF-80 | **1.48e-2** | **2.57e-2** | 7.56e-2 | **2.42e-2** | **2.30e-2** |
| R-LA-160 | 1.40e-2 | 2.42e-2 | 7.16e-2 | 2.29e-2 | 2.37e-2 |
| R-PA-160 | 1.32e-2 | 2.18e-2 | **6.81e-2** | **2.15e-2** | 2.30e-2 |
| R-VF-160 | **1.31e-2** | **2.17e-2** | 6.95e-2 | **2.15e-2** | **2.18e-2** |

Table 3: The results of scaling dataset Microreactor2D. Errors are shown by each component. $L_s$ is sensitivity error. **Bold** is the best and underline is the second best.

| Models | $p$ | $u$ | $v$ | $c$ | $L_s$ |
|---|---|---|---|---|---|
| R-LA | 1.29e-2 $\pm$ 3.3e-4 | 2.33e-2 $\pm$ 1.04e-3 | 6.85e-2 $\pm$ 1.94e-3 | 2.32e-2 $\pm$ 9.3e-4 | 2.16e-2 $\pm$ 3.55e-4 |
| R-PA | 1.25e-2 $\pm$ 4.1e-4 | 2.22e-2 $\pm$ 6.6e-4 | 6.6e-2 $\pm$ 1.23e-3 | 2.28e-2 $\pm$ 2.1e-4 | 2.02e-2 $\pm$ 7.8e-4 |
| R-VF | **1.2e-2** $\pm$ 3.2e-4 | **2.06e-2** $\pm$ 6.1e-4 | **6.42e-2** $\pm$ 1.42e-3 | **2.14e-2** $\pm$ 6.89e-4 | **1.76e-2** $\pm$ 3.81e-4 |

Table 4: Results of Microreactor2D with 5 random seeds. Showing consistent advantage of VF.

