# OpenReview forum: "Accelerating PDE-Constrained Optimization by the Derivative of Neural Operators"
_ICML.cc/2025/Conference — ICML 2025 poster_

### Official Review · Reviewer_kQVE · 2025-02-18

**Overall Recommendation:** 2

**Summary:**

The paper improve on the reference neural operator (RNO) for PDE-constrained optimization (PDECO) by (1)  using the full trajectory data of PDECO and the sensitivity loss (2) introducing the virtual Fourier layer (3) using numerical solution to correct the neural operator solution.

**Claims And Evidence:**

The claims are generally supported by the evidence.

Some clarification is needed regarding Figure 6:
- Why is the objective increasing as the number of calls increases? From the appendix, this example (Microreactor2D) is a minimization problem.
-  As mentioned in the caption,"R-VF w/o reference represents the optimization process relying solely on the R-VF model without any calls to the numerical solver". I think that means the num. of calls should be 0 for R-VF w/o reference? How do we get the blue line

**Essential References Not Discussed:**

The author should discuss how does their method of handling different geometry differs from [1]

And how does their use of derivative differs from [2]

[1] Li, Z., Kovachki, N., Choy, C., Li, B., Kossaifi, J., Otta, S., Nabian, M.A., Stadler, M., Hundt, C., Azizzadenesheli, K., Anandkumar, A., 2023. Geometry-Informed Neural Operator for Large-Scale 3D PDEs.

[2] O’Leary-Roseberry, T., Chen, P., Villa, U., and Ghattas, O. Derivative-informed neural operator: an efficient framework for high-dimensional parametric derivative learning.

**Experimental Designs Or Analyses:**

See Claims And Evidence.

**Methods And Evaluation Criteria:**

The proposed methods make sense. While the evaluation require improvement.

- Table 1 seems to come from 1 testing data set. The performance should be evaluated on multiple testing problems and report the mean/std.
- From table 1, the improvement from VF also seems to be marginal. For example, comparing the best and the second best, the objectives (L) decreases by roughly 0.03, 0.02, 0.02, 0.01 etc. The improvement might not be robust when the number of test case increases.

And some clarification are needed:
- Algorithm 2, what is $u_{gt}$, is it the numerical solution corresponding to the current $\lambda$? Or is it the "ground truth solution to the PDECO"?
- Figure 4, how to make the predicted $u$ and $J$ exact (blue arrows) ? From my understanding of algorithm 2, we don't have access to the red trajectory.

**Other Comments Or Suggestions:**

No other comments.

**Other Strengths And Weaknesses:**

Strength:
- the example problems are challenging.

Weakness:
- see other boxes.

**Questions For Authors:**

Since numerical solutions are used to assist RNO, it's unclear that the hybrid approach is more effective than using numerical solver alone. In particular, the key questions for practitioner will be:
- How many sample/trajectories are needed to generate the training data?
- What is the total time for pre-training and what is the time for each solve?

From figure 4, it seems that the performance stays flat after 4 function calls, this suggest that the accuracy can not be improved with more function calls for Hybrid approach. It's totally reasonable that the accuracy is lower than the numerical solver, but the author need to show that the speed makes up for the loss of accuracy.

Another important question is, does using the sensitivity make the Neural operator less general? An important advantage of NO is that, once pre-trained, it can be used for different objective function. With proposed method, it seems modifying the objective function would require retraining the whole neural operator.


Since RNO is an essential part of the work, I think the author should review RNO with more detail, in addition to the high level introduction in section 3.3. Some related questions:
- Are both $u$ and $\lambda$ are nodal function on a mesh, or $\Omega\rightarrow \mathbb{R}$ Is it really possible to have $\varphi(\lambda_r)=\lambda_q$ and $u_r \circ \varphi ^{-1} = u_q$?
- What is the training pipeline for RNO? How to represent and sample $\varphi$?

**Relation To Broader Scientific Literature:**

The work improve on an existing method (RNO) for PDECO. While effective, the broader impact might be limited.

**Theoretical Claims:**

There is no theoretical claim.

---

> ### Author Rebuttal · Authors · 2025-03-31
>
> Thank you for your review with carefulness and details, and it is highly valuable and helpful to us!
>
> 1.	**Claims and Evidence**
>
> Nice catch on the mismatch of objective! As you pointed out, the objective of Microreactor is to minimize $J = -\int…$ in eq. (17) in appendix C.2. This objective is equivalent to maximize $J=\int…$ without the negative sign. We will revise the form of objectives to be consistent. In our implementation, we used maximization, please see `Line 726 in utils.py`.
> The blue line in Fig. 6 is obtained for illustration purpose. R-VF w/o ref. indeed runs without being given ground truth solution along optimization process. We check the quality of its output in the same way as R-VF with ref. The only difference is to drop the reference GT solution from input.
>
> 2. **Methods And Evaluation Criteria**
>
> We respectfully disagree with the evaluation of marginal improvement. First, let us clarify that in Table 1, errors are shown by each component, and $L_{\cdot}$’s are sensitivity errors. We will modify the caption of Table 1 accordingly. We should emphasize that our main improvement is on derivative learning, which achieves error reduction 3.2%, (1.8%, 2%), (1.2%, 3.5%), 6.1% (or relatively 15.8%, (3.9%, 7.2%), (3.8%, 6.85%), 10.8%) from the second best. Second, the baseline models GNOT and Transolver are SOTA neural operators (NO), which leaves small room for improvement at operator learning (on physical fields). However, our model achieves Pareto front in the tradeoff between operator learning and derivative learning (See Fig. 5 and Section 4.1.1). Please see also **Item 1 for Rebuttal for Reviewer 1** for the reason of the improvement of VF.
>
> We appreciate the rigorousness of the reviewer and decide to supplement some experiment to address the question on the robustness of the improvement. Due to space limit, please check the table in **Item 2 of the rebuttal for Reviewer 2**.
>
> $u_{gt}$ indicates the numerical solution corresponding to the current $\lambda$, NOT the numerical solution of PDECO.
>
> It is a good question on how to make predicted $u$ and $J$ (blue arrow) exact, which is the key to our method.  Given a GT solution ($\lambda_r, u_r$) obtained from numerical solver,  let $\lambda_q =\lambda_r$ and thus $\varphi=0$. The input would be $(\lambda_q, u_r, 0)$. The output of reference neural operators (RNO) should be **close** to the exact reference solution $u_r$ due to our training algorithm. For each query, dataloader would randomly pick neighboring samples from the same optimization trajectory as references, including the queried data itself (See App. C.1). It forces RNO to learn an identity mapping when the deformation $\varphi$ vanishes.
>
> 3.	Reference not discussed.
>
> On geometry handling, [1] belongs to the framework of traditional NO, i.e., learning the mapping between geometry and solution. RNO is a generalization of NO to learn the change of solution according to deformation of geometry (and more general parameters, see Remark 3.2).
>
> [2] is a relevant work, and we thank the reviewer for pointing this out. It has been cited in line 160-161 right. [2] is a general work on training NO in Sobolev space efficiently by exploiting the intrinsic low-dimensionality of the derivatives when handling the high-dimensional Jacobian. Our work learns the directional derivative of NO since the derivative is taken wrt some objective functions, which can be view as a special case of [2].
>
> 4.	Other questions
>
> We apologize for missing the information of dataset sizes. Please see **Item 1 in Rebuttal 2** for details due to space limitation.
>
> Please see **Item 3 in Rebuttal 2** for discussion on wall-clock time for optimization.
>
> It is a great question on if derivative learning would make NO less general. On the problem Inductor2D, we consider a flexible objective with a changing weight $\gamma$ between two terms (See eq. (19)). The dataset is drawn with random sampling of $\gamma$. Therefore, the NOs are able to predict different objectives for different $\gamma$. In principle, the dataset determines the generalization of NO, including objective functions.
>
> Regarding RNO, $u$ and $\lambda$ are nodal functions on mesh. It is indeed possible to have $\varphi$ satisfying the relation between reference and query. To give a simple example, let $\varphi$ be a spatial rotation, and then the nodal value of $u$ remains unchanged, while coordinates $\lambda$ will be rotated accordingly. In the domain of shape optimization, the existence of $\varphi$ is the cornerstone for all applications. See Remark 3.2 for more details.
>
> The training pipeline of RNO is summarized in Algorithm 1 in Appendix D. To represent and construct $\varphi$ would be rather simple with the data from optimization. One can use the shift of mesh nodes between ref. and query for shape optimization, or use the difference between control variable $\lambda$ of ref. and query for topology optimization. See line 256-265 right in the paper.

---

### Official Review · Reviewer_1Qkw · 2025-03-05

**Overall Recommendation:** 2

**Summary:**

The authors propose a neural-operator based approach for solving PDE-constrained optimization problems. While this approach is not new in the literature, the authors come up with several innovations to improve existing approaches: (i) data-driven training by using trajectories generated by traditional optimization algorithms, (ii) enhanced derivative learning through virtual-Fourier layers, and (iii) hybrid optimization integrating neural operators with numerical solvers. The authors show the potential of their approach on several datasets from shape and topology optimization.

## update after rebuttal
I thank the authors for responding to my comments. I have raised my score from 1 -> 2. I am hesitant to raise my score further because of the initial quality of presentation in the paper.

**Claims And Evidence:**

I do not believe that the claims made in the submission are properly supported.

The authors claim in the abstract that their “extensive experimental results demonstrate the effectiveness of our model in accurately learning operators and their derivatives”. However, what is the threshold for accuracy? I am also skeptical of this claim because their method is outperformed by the numerical method in Figure 6. To me this suggests that the derivatives are not being learned to sufficient accuracy. Furthermore, I’m not fully convinced that experiments of four different datasets are enough to be considered extensive.

Also, how exactly do the authors address the data efficiency issues that they describe in the abstract? I think the authors should compare to other data sampling approaches (other than generating data from traditional optimization algorithms) to show that their method yields an improvement.

**Essential References Not Discussed:**

The paper is missing references to DeepONet, which is also popular in the operator learning literature.

There is also work within the last two years that generalizes FNO to irregular meshes, e.g., geo-FNO (https://www.jmlr.org/papers/volume24/23-0064/23-0064.pdf) — the authors should address this in the related work section, where they claim that the applicability of FNO to irregular meshes is limited.

**Experimental Designs Or Analyses:**

The experimental design and analysis seem fine.

**Methods And Evaluation Criteria:**

How much data is needed to train the neural operator in this framework? Is this data easy to generate? If the data generation itself is very expensive, would this method be useful for PDE-constrained optimization?

Another thing the authors should address is the runtime (in seconds) of their method, especially since it seems like the classic numerical method outperforms their approach in Figure 6. Does Hybrid R-VF at least have a faster runtime than the numerical method?

**Other Comments Or Suggestions:**

Using $\hat \cdot$ to denote variables with ground-truth values could be confusing to readers skimming the paper. Typically $\hat \cdot$ is used to denote an estimate of a ground-truth value.

I’d recommend adding a notation subsection at the end of the introduction. For example, the bold notation for tensors and the use of $\mathcal F$ for the Fourier transform could be part of this subsection.

“Softmax” and “Project” are italicized in Remark 3.1, but they are not italicized in the rest of the paper.

**Other Strengths And Weaknesses:**

Combining neural operators with PDE-constrained optimization is an interesting idea. However, there is a lot of room for improvement in the presentation (see “Other Comments Or Suggestions” and “Questions For Authors”). I found section 3.2, which introduces virtual-Fourier layers, particularly hard to follow. I had a hard time distinguishing what aspects of the virtual-Fourier layer were being introduced by the work, and what aspects were just based off of the FNO.

**Questions For Authors:**

1. Page 2: The paper refers to “sequences of solutions and sensitivities” that are computed by the adjoint method. However, the mathematical expressions for the sensitivities are never shown in the paper. Presenting the mathematical expressions for these sensitivities (and perhaps, a brief overview of the adjoint method) will improve the quality of presentation in the paper.

2. Page 4: What does it mean for the Fourier transform of Fourier-based layers to be “practically linear”. Does this mean that the Fourier transform of these layers is a linear operator, or that it has linear time complexity. This statement should be clarified in the paper.

3. Page 4: What does it mean for the derivative to “not add additional bias to the layer”? Is this supposed to motivate why the authors use FNO rather than transformer-based architectures?

4. Page 5: After the discussion of virtual-Fourier layers in section 3.2, the paper abruptly jumps to training and optimization with RNO. How are the virtual-Fourier layers related to the RNO? Are they the building block of RNO?

**Relation To Broader Scientific Literature:**

To the best of my knowledge, the paper makes an interesting contribution to the broader literature on PDE-constrained optimization. In particular, the virtual-Fourier layer seems to be novel, and could be used in future work to improve the derivative accuracy in neural operators.

**Theoretical Claims:**

There are no theoretical claims made in the paper.

---

> ### Author Rebuttal · Authors · 2025-03-31
>
> Thank you for reviewing our paper, your feedback is in-detail and comprehensive. Among your comments we notice that you particularly expressed “having a hard time following virtual-Fourier (VF) layers and distinguishing VF from FNO”. We would like to clarify this point first since it lies in the core of our motivation of this work.
>
> FNO and its variants require a uniform-grid domain due to the need of applying fast Fourier transform. geo-FNO uses a 1-to-1 mapping to deform a uniform grid to an arbitrary shape, and the computation essentially still happens on uniform grids. To simply put, the input shape (H, W, C) must be fixed. VF on the other hand can be applied to arbitrary number of mesh grids (T, C), where the length of input T can change. This is similar to transformer-based neural operators and preferable since we are dealing with data from shape/topology optimization, where **mesh grids can be arbitrary and uniform grids are inapplicable.**
>
> The way of handling arbitrary length input with VF layer consists of, 1. Project the input onto a **fixed** number of virtual sensors; 2. Process the virtual signals by Fourier layers; 3. Project the processed signals back to physical space (See Figure 3). The projection technique is entirely independent from FNO and inspired by Transolver instead. Additionally, our projection is different from Transolver due to our consideration of learning derivatives (See Remark 3.1).
>
> We emphasize that, to our best knowledge, this is the first work that extends Fourier layers to arbitrary mesh grids. The benefit is its advantage in learning derivatives due to its simple structure compared to transformer counterparts (See items 2&3 below).
>
> To address your other questions:
>
> 1.	Sensitivity is defined on line 69 left by equation (2),  $\frac{\partial \tilde{J}}{\partial \lambda}  = \frac{\partial J}{\partial u} \frac{\partial u}{\partial \lambda} + \frac{\partial J}{\partial \lambda}$. We will bold the word sensitivity here.
>
> 2.	We meant Fourier transform is a linear operator, which has a simple derivative, i.e., $\frac{d}{dx}\mathcal{F}(z)= \mathcal{F}(\frac{dz}{dx})$. This motivates us to integrate Fourier layer into our design to learn derivatives. We will modify this part to improve clarity.
>
> 3.	Consider the derivative of the attention of a transformer, $\frac{d}{dx} softmax(QK)V$. The derivative must be taken wrt to Q, K, V, hence involving product rule. Also, the derivative of softmax is $s_i(\delta_{ij}-s_j)$ (See line 189-191). Thus the derivative of attention would be roughly and informally $s_i(\delta_{ij}-s_j)(Q’K+QK’)V + S(QK)V’$. Hence, we pointed out that the derivative of transformer is assigned with strong inductive bias (probably undesired). Imagine learning derivatives of operators with layers constructed as above. The expressiveness is therefore restricted.
>
> On the other hand, the derivative of Fourier transform is as simple as $\frac{d}{dx}\mathcal{F}(z)= \mathcal{F}(\frac{dz}{dx})$, and hence has “no additional bias”. We conducted more analysis of the derivatives of VF in Remark 3.1 and Appendix A.
>
> 4.	Indeed, VF is the building block of RNO. Namely, according to the general architecture of neural operators $\mathcal{G}_{\theta}:= \mathcal{Q}\circ\mathcal{L}\circ\cdots\circ\mathcal{L}\circ\mathcal{P}$ (On line 149 right, in the beginning of Section 3), VF instantiates the integral operator $\mathcal{L}$. To smoothly transit between sections, we will clarify the connection in the beginning of section 3.3.
>
> 5.	**Claims and Evidence**
>
> The reason why we did not compare with traditional sampling method is the following. For example, consider random sampling in parametric space of the drone problem (Fig. 10). The parameter space is $[0,1]^{30000}$, given 3E4 mesh nodes. The likelihood of sampling an optimal design is almost surely zero. Therefore, an optimal design would be an out-of-distribution (OoD) data for random sampling method. Since neural operators would fail on predicting OoD data, they are inapplicable for optimization tasks. Thus, in order to make data near optimality in-distribution, in our opinion, it is necessary to adopt optimization data sampling. See the discussion on line 97-109 left in our paper.
>
> The data efficiency of our approach is exemplified in Table 1. Due to space limit, please see discussion in **Item 1 of Rebuttal 2**.
>
> 6.	**Methods And Evaluation Criteria**
>
> We apologize for missing the detail of data size in our submission. Please also see **Item 1 of Rebuttal 2** for details. The dataset size we use is around 1E3, which is similar to the setup of many traditional neural operators. Therefore, the cost of dataset is NOT significantly more than traditional datasets.
>
> Regarding runtime, please see discussion in **Item 3 of Rebuttal 2**.

---

### Official Review · Reviewer_V7mT · 2025-03-09

**Overall Recommendation:** 4

**Summary:**

This paper proposed a novel framework to enhance PDE-constrained optimization (PDECO) with neural operators, addressing data efficiency and robustness. Key innovations include data-driven training, a Virtual-Fourier layer for improved derivative learning, and a hybrid optimization approach integrating neural operators with numerical solvers. Experiments show accurate operator learning, robust convergence, and improved optimization performance.

**Claims And Evidence:**

Yes.

**Essential References Not Discussed:**

The authors may add some reviews of papers from 'AI-aided geometric design of anti-infection catheters' and 'Physical Design using Differentiable Learned Simulators'

**Experimental Designs Or Analyses:**

All Look good to me.

**Methods And Evaluation Criteria:**

Looks good to me.

**Other Comments Or Suggestions:**

No such.

**Other Strengths And Weaknesses:**

No such.

**Questions For Authors:**

No such.

**Relation To Broader Scientific Literature:**

We need such optimization works using operator learning.

**Theoretical Claims:**

No such

---

> ### Author Rebuttal · Authors · 2025-03-31
>
> Thank you for reviewing our paper, your summarization is precise and your suggestion on additional references are well received. Particularly, we thank you for suggesting the 2nd work that we'd like to discuss the distinction with our work. The first distinction is that their work is targeting time-dependent fluid problems. Our work is targeting static problems, e.g., steady flow, electromagnetics on frequency domain, solid mechanics. Second, their approach intends to replace numerical simulation with surrogate models. Our work is complementary to surrogate approach that hybrids with numerical solvers. It is an interesting question on how to apply our hybrid method to problems with rollout operations, which can be explored in the future.
>
> Below are some frequently asked points we’d like to address:
>
> 1.	Dataset sizes and Table 1
>
> Microreactor, Fuelcell, Inductor, Drone have 100x12, 30x20, 100x10, 10x20 samples respectively as in num_trajectory x num_steps. We added the information in appendix.
>
> In Table 1, all errors are listed by components, and $L_{\cdot}$’s are sensitivity errors (we will clarify this in caption). Naïve supervised learning paradigm for neural operators perform poorly due to overfitting highly correlated optimization data. The data from optimization steps are quite similar to each other.  Our solution to learn by reference neural operators (RNO) is much more efficient in terms of low error rate (Comparing models with and without “R-”).
>
> To see the improvement due to virtual-Fourier (VF), we observe that the sensitivity errors are reduced by 3.2%, (1.8%, 2%), (1.2%, 3.5%), 6.1% (or relatively 15.8%, (3.9%, 7.2%), (3.8%, 6.85%), 10.8%) from the second best. See also the balance of tradeoff between error and sensitivity error in section 4.1.1, VF achieves the best Pareto front.
>
> 2.	Robustness of improvement of R-VF
>
> The following results are obtained on 4 runs on Microreactor with different random seeds. R-VF consistently outperforms the baselines. Note that, LA: Linear Attention, PA: Physics Attention, and R-: reference neural operators.
> | Models | p | u | v | c| $L_s$ |
> |---|---|---|---|---|---|
> |R-LA| $1.29e-2 \pm 3.3e-4 $ | $2.33e-2  \pm 1.04e-3 $ | $6.85e-2 \pm 1.94e-3$ | $2.32e-2 \pm 9.3e-4$ | $2.16e-2 \pm 3.55e-4$ |
> | R-PA | $1.25e-2 \pm 4.1e-4 $ | $2.22e-2  \pm 6.6e-4 $ | $6.6e-2 \pm 1.23e-3$ | $2.28e-2 \pm 2.1e-4$ | $2.02e-2 \pm 7.8e-4$ |
> | R-VF | $\textbf{1.2e-2} \pm 3.2e-4 $ | $\textbf{2.06e-2}  \pm 6.1e-4 $ | $\textbf{6.42e-2} \pm 1.42e-3$ | $\textbf{2.14e-2} \pm 6.89e-4$ | $\textbf{1.76e-2} \pm 3.81e-4$ |
>
> Also, we doubled the size of Microreactor dataset to 200 trajectories (each of 12 steps and effectively 2400 samples). 40 traj. are kept as test set. We train models on 80, 160 traj. respectively.  We observe that with larger dataset, R-VF kept advantage over baselines on predicting both physical fields and sensitivities. The main improvement is still on sensitivities.
> |     | p | u | v | c| $L_s$|
> |---|---|---|---|---|---|
> |R-LA-80| 1.64e-2 | 3.25e-2 | 7.83e-2 | 2.65e-2 | 2. 69e-2 |
> | R-PA-80 | 1.51e-2 | 2.74e-2 | $\textbf{7.46e-2}$ | 2.48e-2 | 2.54e-2 |
> | R-VF-80 | $\textbf{1.48e-2}$ | $\textbf{2.57e-2}$ | 7.56e-2 | $\textbf{2.42e-2}$ | $\textbf{2.30e-2}$ |
> |R-LA-160| 1.40e-2 | 2.42e-2 | 7.16e-2 | 2.29e-2 | 2.37e-2 |
> | R-PA-160 | 1.32e-2 | 2.18e-2 | $\textbf{6.81e-2}$ | $\textbf{2.15e-2}$ | 2.30e-2 |
> | R-VF-160 | $\textbf{1.31e-2}$ | $\textbf{2.17e-2}$ | 6.95e-2 | $\textbf{2.15e-2}$ | $\textbf{2.18e-2}$ |
>
>
> 3.	Wall-clock time for optimization with R-VF
>
> The runtime of our hybrid method consists of time on optimizing with NO and time on calling numerical solvers. The latter dominates the runtime, and optimization with NO is much cheaper. A potential cost saving of hybrid method lies in fewer calls of numerical solvers.
>
> One crucial aspect of optimization with neural operators is the method of optimization. In our implementation we adopted GD and MMA, but there are many other choices such as Adam, L-BFGS, SIMP, etc. A good optimization method can reach higher objective values and save both the numbers of iterations and function calls. Thus, the optimization method significantly affects wall-clock time. Besides, there are some important techniques in optimization, e.g., gradient filtering, projection, clipping, etc. A comprehensive and fair comparison requires all these techniques. Therefore, in our opinion it is premature to compare the runtime in this work.

---

> > ### Comment · Reviewer_V7mT · 2025-04-04
> >
> > The derivative of Neural Operator in general is of interest and importance.

---

> > > ### Author Response · Authors · 2025-04-04
> > >
> > > Thank you again for reviewing our paper. Your comments on our contribution are precise and concise. Your suggestion on reference is valuable and helpful.

---

### Official Review · Reviewer_wq4G · 2025-03-14

**Overall Recommendation:** 4

**Summary:**

Authors propose a general architecture-agnostic framework that can be used for PDE-constrained optimization and a Virtual Fourier Layer suitable for data on irregular grids.

The framework consists of three essential parts: (i) a particular structure of inputs and outputs for selected neural networks, (ii) a way to generate training data and train the model, (iii) an inference strategy that utilises classical solver.

(i) Roughly, the inputs to the neural networks are (solution for current parameters, desired parameters parameters, difference between current and desired parameters) and the output is a solution for desired parameters.

(ii) Data is generated along the trajectories of the optimization process and the training is performed with intermittent masking of two inputs: (solution for current parameters, difference between current and desired parameters). The latter encourages neural networks to be more robust and learn mapping from parameters to solution.

(iii) At the inference stage authors propose to sparingly correct optimisation trajectory using classical solver. This is possible because of the special structure of the neural network described in (i).

## update after rebuttal

Summarised in https://openreview.net/forum?id=LFF7kUQ5Rp&noteId=gR2VRvNh9Z

**Claims And Evidence:**

In my view most claims made by authors are well-supported.

The one missing metric is a wall-clock time, required by different methods to reach optimal solution or approximation to thereof. Without this data it is not possible to claim that the proposed method leads to faster PDE-constrained optimization as stated in the title.

**Essential References Not Discussed:**

Authors do not discuss development of surrogate modelling at all. In place of that they provide references for related problems: prediction with neural operators, training that promotes smoothness, hybrid solvers that combine neural networks and classical techniques.

I suggest authors provide a brief review of PDE-constrained optimization methods that incorporate neural networks for surrogate modelling. Several references are provided above, but more thorough review will improve the overall quality of the contribution.

**Experimental Designs Or Analyses:**

The most important part of experiments is the ablation study done by authors.

The proposed approach consists of three techniques: (i) training strategy and the overall design of architecture (inputs and output), (ii) Virtual Fourier Layer, (iii) inference strategy. Given that, it is reasonable to evaluate the effect of each component.

Quite appropriately authors perform such ablation tests and present the results in Table 1. From this table I can conclude that corrections with reference at inference play the most significant role, the effect of training strategy is the second most important factor and the improvement from the use of Virtual Fourier Layer is the least important but still pronounced.

**Methods And Evaluation Criteria:**

Authors consider several challenging PDE-constrained optimisation problems and perform reasonable ablation study. I believe that both methods and evaluation criteria are adequate.

**Other Comments Or Suggestions:**

The article is generally well-written, so I have only a few minor points:
1. The need for Figure 1 is not evident, since it does not explain much beyond the fact that optimization trajectory with approximate solver diverges from trajectory with more accurate solver. Figure 4 seems to demonstrate the same effect.
2. Line 408, right column. "without numerical solver feedback quickly derails." -> "Without numerical solver feedback quickly derails."

**Other Strengths And Weaknesses:**

**Strengths:**
1. The approach by authors is clearly more robust than alternative. This robustness is achieved by learning small corrections to the reference solution and by the use of classical solvers that can provide unbiased reference.
2. The developed framework is general and can be used applied with arbitrary architectures

**Weaknesses:**
1. Classical solver is still required at the inference stage.
2. More challenging collection of training data: (i) data is collected along optimisation trajectories, (ii) in addition to input-output pairs the derivative of the loss with respect to parameters is collected.

**Questions For Authors:**

1. Section 3.2. contains several fragments on bias and complexity of derivatives. For example:
   1. Lines 187-188 "For a transformer, the nonlinearity of attention unit causes complex calculation of derivatives ..."
   2. Lines 196-197 "Fourier-based layers is practically linear, and therefore does not introduce additional biases ..."
   3. Lines 190-192, right column "The derivative of equation (8) introduces less bias due to its simpler structure, which motivates us to adopt this form."
   4. Lines 210-213 "Since Fourier-based layer is practically linear, its derivative does not add additional bias to the layer. This is highly favorable for derivative learning."
   I do not understand these claims. Can the authors please clarify what they mean by the "complex calculations" of derivatives? Do they refer to numerical complexity? Isn't it the case that numerical complexity of automatic differentiation (AD) is roughly the same as forward pass through the architecture?
   Next, what do authors mean by "bias" of derivative? Again, since AD is used the derivative is computed up to machine precision for all but pathological cases. Since authors use fairly stable architectures and compute derivatives with respect to input, I would not suspect to see instabilities in AD.
2. In equation (12) authors use $\varphi$ and explain that for topological optimization this is $\lambda_r - \lambda_q$. From Appendix one can find that this is the case for all considered examples. Can the authors please provide an example where $\varphi$ has a different form or reflect in the main text that $\varphi$ is a difference of controlled variables for most of the problems.

**Relation To Broader Scientific Literature:**

PDE-constrained optimization is a mature field with a large number of classical techniques available https://link.springer.com/book/10.1007/978-3-319-13395-9.

The approach by authors is within a scope of surrogate modelling https://arc.aiaa.org/doi/10.2514/6.2000-4891. In the PDE-constrained optimization a repeated solution of PDE is required, so the idea is to replace accurate but numerically expensive classical solver with numerically cheap surrogate model. Various surrogate models were developed including ROM https://link.springer.com/chapter/10.1007/978-3-642-55508-4_16 and neural networks https://arxiv.org/abs/2111.04941, https://arxiv.org/abs/2110.13297.

The main innovation of the current contribution is a design of a special training scheme and network structure that allows one to use a numerical solver in conjunction with a surrogate model.

**Theoretical Claims:**

Authors do not provide novel theoretical claims.

---

> ### Author Rebuttal · Authors · 2025-03-31
>
> Thank you for reviewing our paper in such depth and being very informative, especially on reference. It is highly valuable to us!
>
> 1.	**Experimental Designs Or Analyses**
>
> We respectfully disagree that Virtual Fourier (VF) Layer is least important. It reflects our thoughts on designing architecture for learning derivatives. To this sense, it is as crucial as inference with reference in our framework.  In Table 1, all errors are listed by components, and $L_{cdot}$’s are sensitivity errors (we will clarify this in caption). We observe that the sensitivity errors are reduced by 3.2%, (1.8%, 2%), (1.2%, 3.5%), 6.1% (or relatively 15.8%, (3.9%, 7.2%), (3.8%, 6.85%), 10.8%) from the second best. This improvement is due to the architecture of VF. See also the balance of tradeoff between error and sensitivity error in section 4.1.1, VF achieves the best Pareto front.
>
> Specifically, the derivative of VF layer has a simpler form than baseline models such as transformers. Consider the derivative of the attention of a transformer, $\frac{d}{dx} softmax(QK)V$. The derivative must be taken wrt to Q, K, V, hence involving product rule. Also, the derivative of softmax is $s_i(\delta_{ij}-s_j)$ (See line 189-191). Thus the derivative of attention would be roughly and informally $s_i(\delta_{ij}-s_j)(Q’K+QK’)V + S(QK)V’$. Hence, we pointed out that the derivative of transformer is assigned with strong inductive bias (probably undesired). Imagine learning derivatives of operators with layers constructed as above. The expressiveness is therefore restricted.
>
> On the other hand, the derivative of Fourier transform is as simple as $\frac{d}{dx}\mathcal{F}(z)= \mathcal{F}(\frac{dz}{dx})$, and hence has “no additional bias”. We conducted more analysis of the derivatives of VF at Remark 3.1 and Appendix A.
>
> 2.	**Questions For Authors**
>
> Following the analysis from last question, we hope that it is clearer to see the derivative of VF having more expressiveness compared to transformer counterparts. We will modify the paper to reduce the ambiguity on “complex derivatives” and “additional bias”.
>
> In our paper, there is another type of example of $\varphi$ besides topology optimization, namely shape optimization. Let $\varphi$ be a deformation between two shapes, which is instantiated as mesh grids warping and finally discretized as mesh grids shift. See Fuelcell and Inductor examples in our experiment. Thus, in this paper, for both topology and shape optimization, the discrete $\varphi$ can be represented as $\lambda_r-\lambda_q$.
>
> It is definitely an interesting question to consider other form of $\varphi$. In fact, the deformation $\varphi$ is not unique. For example, given two shapes of the same topology, there are infinite many ways of deforming one to another.  How does $\varphi$ affect the operator learning? Is it possible to take integral of RNO along a path of warping a shape to a much more different shape beyond perturbing? We believe there is a vast field to explore in this area.
>
> 3.	**Weakness**
>
> We appreciate your precise judgement on recognizing the reliance of our method on classic solver and the nature of data as optimization trajectories. However, we’d like to discuss with you as these topics do not present as weakness in our opinion.
>
> Hybrid methods have great potential in scientific computing area. One of the main reasons is that modern computing algorithms are developed with profound theories and abundant ecosystems, e.g., open/closed source software. The guaranteed accuracy and reliability are still indispensable in the era of data-driven approach. We are glad you mentioned the work and effort of surrogate models. Advocates embrace them for the dominating advantage in cost. We absolutely agree and would love to contribute to this community. However, we hold a slightly different view that AI will not replace numerical computing but rather play as a complementary part, which will be different from what happens to CV or NLP. It is because of this reason hybrid method is getting popular in AI4PDE, e.g., Multigrid method and other hybrid methods mentioned in our paper.
>
> The cost of optimization dataset is NOT larger than traditional neural operators. We apologize for missing this information of dataset sizes and added to appendix. Please see **item 1 in Rebuttal 2** for details due to space limit. Also, in our opinion the derivative information would be necessary if we tend to obtain surrogate models with good gradient property.
>
> 4.	Missing metric of wall-clock time
>
> Please see **Item 3 in Rebuttal 2** due to space limit.

---

> > ### Comment · Reviewer_wq4G · 2025-04-02
> >
> > In my view the techniques proposed by authors are valuable and reasonable. Besides that the claims are well supported numerically. I revise my recommendation accordingly.
> >
> > I read the comment on computation time required for different methods and I agree that it is not straightforward to perform a fair comparison. My suggestion for the author is to include this discussion in the main body of the paper and provide at least some reference running times for the readers.

---

> > > ### Author Response · Authors · 2025-04-03
> > >
> > > Thank you for your quick reply and being such a responsible reviewer! We warmly agree with your suggestion, the discussion on runtime will be included in the main body.
> > >
> > > Below we summarize the runtime of optimization based on RNO and numerical method. We run the experiment both on a laptop with CPU 2.5 GHz (11th Gen Intel i7) and on GPU Nvidia V100. The time is an average value on 10 iterations. Unit is second per iteration. The case of Microreactor has 4.9E3 mesh nodes, and the case of Drone has 2.1E4 mesh nodes.
> > >
> > > |     | R-VF (GPU) | R-VF (CPU) | Numerical (CPU) |
> > > |---|---|---|---|
> > > |Microreactor2D| 0.53 | 1.89 | 3.20 |
> > > |Drone3D| 0.62 | 3.20 |49.40 |
> > >
> > > The runtime of numerical method severely increases as the problem scales. This is because R-VF has a linear complexity wrt number of mesh nodes, and numerical method is known to suffer from curse of dimensionality.
> > >
> > >
> > > We sincerely thank you again for your overall reviews, your expertise has truly helped us improving the comprehensiveness, clarity and rigorousness.

---

### Decision · Program_Chairs · 2025-05-01

**Decision:**

Accept (poster)

**Comment:**

This submission considers a novel neural-operator approach (improving on the reference neural operator approach) for PDE-constrained optimization (PDECO). The main ideas are a virtual-Fourier layer for learning derivatives and improved training by modifying the optimization scheme. In my opinion, the authors have sufficiently addressed the majority of the concerns brought up by reviewers during the rebuttal period (and scores have been raised as a result, giving final scores 4,4,2,2). Although there are still some concerns about presentation (e.g., choices of notation, convoluted explanations), overall the reviewers agree that the work is novel and interesting to the community. I therefore recommend to accept this paper and strongly encourage the authors to work on the presentation aspects of the paper.